# Effects of Endurance and Resistance Training on Cardiovascular Outcomes and Quality of Life in Patients with Heart Failure with Reduced Ejection Fraction: A Structured Narrative Review

**DOI:** 10.3390/jfmk10040483

**Published:** 2025-12-18

**Authors:** Michael Stiefel, Jamie O’Driscoll, Hadassa Brito da Silva, Tristan Ramcharan, Michael Papadakis

**Affiliations:** 1Department of Cardiology, University Heart Center, University Hospital Zurich, 8091 Zurich, Switzerland; michael.stiefel@usz.ch; 2Leicester Lifestyle and Health Research Group, Leicester Diabetes Centre, University of Leicester, Leicester LE5 4PW, UK; jod16@leicester.ac.uk; 3Hochgebirgsklinik Davos, Medicine Campus Davos, 7265 Davos, Switzerland; hadassa.britodasilva@hgk.ch; 4Cardiovascular Clinical Academic Group, City St George’s, University of London, London EC1V 0HB, UK; tristanramcharan@doctors.org.uk; 5Heart Unit, Birmingham Children’s Hospital, Birmingham B4 6NH, UK

**Keywords:** exercise training, cardiac rehabilitation, VO_2_peak, ventilatory efficiency (VE/VCO_2_), quality of life (QoL)

## Abstract

**Background:** Heart failure with reduced ejection fraction (HFrEF) markedly impairs quality of life (QoL) and life expectancy. The main therapeutic goals are to reduce mortality, improve functional capacity, and enhance QoL. Exercise training is an evidence-based, non-pharmacological component of standard care that improves functional capacity and clinical outcomes in HFrEF. This review examines the effects of endurance and resistance training on peak oxygen uptake (VO_2_peak), ventilatory efficiency (VE/VCO_2_ slope), health-related QoL, and cardiovascular outcomes in patients with HFrEF. **Methods**: A structured narrative review was conducted using comprehensive searches of PubMed, EMBASE, and the Cochrane Library for English-language studies published between January 2004 and October 2024. Eligible studies included adult HFrEF populations undergoing aerobic and/or resistance training with reported effects on VO_2_peak, ventilatory efficiency, QoL, or clinical outcomes. Given the heterogeneity of interventions, comparators, and outcome metrics, data were synthesized descriptively. **Results**: Across 18 studies (plus one sub-analysis) including 3401 patients, 17 trials assessed VO_2_peak and 16 reported significant improvements, with an average increase of approximately 2 mL·kg^−1^·min^−1^. Six studies assessed ventilatory efficiency, and five demonstrated reductions in VE/VCO_2_ slope averaging 4.4 units. Eleven studies analyzed QoL, and nine reported significant improvements corresponding to an ≈5-point decrease in the Minnesota Living with Heart Failure Questionnaire (MLHFQ). In the largest trial, exercise training was associated with modest but statistically significant reductions in all-cause mortality or hospitalization (HR 0.89) and cardiovascular mortality or heart-failure hospitalization (HR 0.85) after adjustment for baseline prognostic factors. **Conclusions**: Structured exercise training improves aerobic capacity, ventilatory efficiency, and QoL in patients with HFrEF, with supportive evidence for reduced morbidity and mortality. These findings underscore the value of structured exercise as a core component of modern HFrEF management.

## 1. Introduction

Heart failure with reduced ejection fraction (HFrEF) is a prevalent condition that substantially impairs quality of life (QoL). Globally, heart failure (HF) affects an estimated 56 million people, with prevalence rates ranging from 1 to 3% in the general population and rising sharply with age. Although age-adjusted incidence has declined modestly in some high-income regions, the absolute burden of HF continues to increase due to population aging and the growing prevalence of cardiometabolic risk factors. Approximately half of all individuals with HF have HFrEF, which occurs more frequently in men, whereas HF with preserved ejection fraction (HFpEF) is more common in women [1,2,3,4].

Coronary artery disease (CAD) and hypertension are the leading causes of HFrEF. Other relevant etiologies include valvular heart disease, genetic and acquired cardiomyopathies, myocarditis, and cardiotoxic agents such as alcohol or chemotherapeutics [5]. While treatment advancements have improved prognosis, mortality and QoL remain concerning [6,7,8]. Women generally have better survival rates [9]. Patients often face frequent hospitalizations, averaging once per year, many due to non-cardiovascular causes [10]. Recent data suggest increased admissions driven by diabetes, obesity, and impaired kidney function [11]. As the population ages and comorbidities rise, hospitalizations for HFrEF are expected to increase significantly [1].

Pharmacotherapy is essential for treating HFrEF, prioritized alongside non-pharmacological interventions before device therapy. Guideline-directed medical therapy includes an angiotensin-converting enzyme inhibitor (ACEi), or an angiotensin II receptor blocker (ARB), or preferably an angiotensin receptor-neprilysin inhibitor (ARNI), together with a beta-blocker, a mineralocorticoid receptor antagonist (MRA), and a sodium-glucose cotransporter-2 inhibitor (SGLT2i). These therapies collectively reduce neurohormonal activation, improve cardiac remodeling, and substantially lower mortality and hospitalization risk. The main treatment goals are reducing mortality, preventing hospitalizations, and improving the functional capacity and QoL [1]. Structured exercise training is the most evidence-based non-pharmacological strategy for improving both physiological and patient-centered outcomes in HFrEF [1,12]. In the general population, broad evidence supports reduced mortality risk through regular physical activity [13,14,15]. Regular physical activity also improves QoL and favorably modifies cardiovascular risk factors [16,17].

Cardiorespiratory fitness (CRF) is strongly and inversely associated with mortality across age, sex, and racial groups, with low fitness conferring greater risk than any single cardiac risk factor [18,19,20]. Aerobic capacity (VO_2_peak) is the most accurate marker of CRF [20], and both VO_2_peak and ventilatory efficiency (VE/VCO_2_ slope) are independent prognostic indicators in HF [21,22]. Chronic HF impairs oxygen transport pathways, leading to exercise intolerance and reduced QoL. Structured exercise training mitigates these deficits by enhancing skeletal-muscle oxygen extraction and metabolic efficiency, thereby improving aerobic capacity and functional performance. Higher VO_2_peak also reflects more efficient muscle metabolism and vascular function, which supports better exercise tolerance and QoL [23].

Earlier meta-analyses have consistently demonstrated beneficial effects of exercise in HF. Gomes-Neto et al. (2019) reported significant improvements in strength and QoL and modest increases in VO_2_peak (+2.9 mL·kg^−1^·min^−1^) in patients with HFrEF, but no consistent changes in ventilatory efficiency [24]. Edwards and O’Driscoll (2022), who evaluated both HFpEF and HFrEF, found improvements in VO_2_peak, exercise tolerance, ventricular function, and QoL, but no significant effects on hospitalization or mortality [25].

These analyses provided valuable quantitative summaries but did not explore the underlying physiological and mechanistic pathways linking exercise adaptations to clinical prognosis [24,25].

Despite robust evidence supporting exercise in HF, uncertainty remains regarding the comparative effectiveness of specific training modalities, variability across patient subgroups, and the extent to which physiological improvements translate into clinical outcomes. Optimal exercise prescriptions for different HFrEF phenotypes also remain incompletely defined. Therefore, the central question of this review is how endurance and resistance training influence aerobic capacity, ventilatory efficiency, QoL, and clinical outcomes in patients with HFrEF. We focused on VO_2_peak, VE/VCO_2_ slope, QoL, and clinical events, as these outcomes represent the most robust and consistently reported prognostic markers in contemporary HFrEF exercise trials.

This structured narrative review synthesizes contemporary trials conducted within the modern era of HFrEF management, integrating recent evidence on aerobic and resistance training to clarify their effects on functional capacity, ventilatory efficiency, and patient-centered outcomes. By drawing on findings from recent randomized and observational studies, it highlights the central role of structured exercise and cardiac rehabilitation in HFrEF care and identifies characteristics of exercise interventions that may offer the greatest prognostic and QoL benefits. These insights may support more personalized and effective exercise prescriptions in clinical practice.

## 2. Methods

### 2.1. Search Strategy

Comprehensive searches were conducted in PubMed, EMBASE, and the Cochrane Library for studies published from January 2004 to 6 October 2024. These databases were selected for their broad and complementary coverage of biomedical, cardiovascular, and rehabilitation research and are commonly used in exercise-cardiology reviews. The 20-year time window was chosen to capture research conducted within the contemporary era of HF management, reflecting modern pharmacotherapy and exercise-rehabilitation practices.

Searches were limited to English-language publications due to resource and translation constraints inherent to a structured narrative review; therefore, some non-English-language studies may not have been identified. Reference lists of prior reviews and included studies were screened manually; however, the review was not designed to re-evaluate all primary studies included in earlier meta-analyses. The aim was to identify contemporary trials reporting one or more of four key outcomes—VO_2_peak, ventilatory efficiency, QoL, and clinical events—within modern HFrEF management. The grey literature (trial registries, conference abstracts, unpublished data) was not systematically searched.

The same search strategy was applied across all three databases (PubMed, EMBASE, Cochrane Library). The identical Boolean search string used in all databases was as follows: (‘endurance training’ OR ‘strength training’ OR ‘exercise’ OR ‘rehabilitation’) AND (‘cardiovascular outcomes’ OR ‘VO2’ OR ‘quality of life’) AND ‘heart failure’ AND ‘reduced ejection fraction’.

### 2.2. Inclusion Criteria

Studies were eligible for inclusion if they

-involved adults (≥18 years) with HFrEF (LVEF ≤ 40–45%);-investigated aerobic and/or resistance training, alone or in combination, compared with usual care or standard therapy;-reported outcomes related to VO_2_peak, VE/VCO_2_ slope, QoL, or clinical endpoints (e.g., hospitalization or mortality).

Randomized controlled and observational studies were both included to provide a comprehensive overview of intervention effects.

### 2.3. Study Selection and Data Extraction

Study screening and selection were conducted using Rayyan (Rayyan.ai, Rayyan Systems Inc., Cambridge, MA, USA, accessed October 2024). From each included study, data were extracted on study design, sample size, baseline characteristics, intervention details (type, intensity, and duration), comparator group, outcomes assessed (VO_2_peak, VE/VCO_2_ slope, QoL, and mortality), follow-up duration, and main statistical findings. Extraction followed a standardized template to ensure consistency across studies. No authors were contacted for additional data.

After removing duplicates, 129 records were screened by title and abstract. Eighty-five were excluded as not relevant to the research question or not meeting inclusion criteria. The remaining 44 articles were assessed in full text; 25 were excluded (most commonly for wrong population, intervention, outcome, or study design). Nineteen studies were included in the final review (Figure 1). Title, abstract, and full-text screening were conducted by the first author, and the co-authors verified the final selection for accuracy and consistency.

### 2.4. Synthesis

Substantial diversity in study design, patient characteristics, intervention modalities (aerobic, resistance, combined, or interval training), and training intensity and duration precluded a quantitative meta-analysis. Consequently, findings were summarized descriptively and grouped according to the primary outcomes of interest (VO_2_peak, VE/VCO_2_ slope, and QoL). This structured narrative framework allowed the integration of physiological, clinical, and mechanistic evidence to clarify how exercise training influences prognosis in HFrEF.

## 3. Results

### 3.1. Study Selection and Overview

Systematic searches of PubMed, EMBASE, and the Cochrane Library yielded 175 records. After removal of duplicates, 129 articles were screened by title and abstract, and 44 were assessed in full text. Eighteen studies (plus one sub-analysis) met the inclusion criteria and examined the effects of exercise training on VO_2_peak, ventilatory efficiency, and/or QoL in adults with HFrEF (Figure 1). Altogether, the included trials enrolled 3401 participants published between 2004 and 2024. Most studies were randomized controlled trials (*n* = 16) and two were prospective cohort studies. Participants generally had an LVEF of 30–40% and NYHA class II–IV (Table 1). Detailed narrative summaries of all included studies are presented in the Appendix A.

### 3.2. Study and Intervention Heterogeneity

Considerable heterogeneity was observed across the included studies. Sample sizes ranged from 19 to 2331 participants, and exercise protocols differed markedly in content, intensity, duration, and supervision. Interventions included center-based programs, hybrid supervised and home-based models, with program durations between 7 and 24 weeks. Exercise frequency ranged from two to five sessions per week, session duration from 20 to 60 min, and intensity from 40 to 90% of VO_2_peak or heart-rate reserve.

The modalities comprised moderate continuous training (MCT), high-intensity interval training (HIIT), combined endurance and resistance programs, and progressive strength or functional programs. HIIT protocols were typically conducted for 12–16 weeks, MCT for 12–24 weeks, and combined or functional programs for 8–16 weeks.

Overall, the studies differed in participant age, HF severity, training dose, and outcome definitions—factors that contribute to variability in reported effects.

### 3.3. Summary of Main Outcomes

#### 3.3.1. Cardiorespiratory Fitness (VO_2_peak)

Seventeen of the nineteen included studies reported VO_2_peak outcomes, and sixteen demonstrated significant improvements. Across trials, VO_2_peak increased by +0.6 to +3.8 mL·kg^−1^·min^−1^ after 7–24 weeks of training, with a median gain of approximately +2.0 mL·kg^−1^·min^−1^ (Table 1).

**Aerobic endurance training**: In the largest trial (HF-ACTION, O’Connor et al. [26]), 2331 patients with chronic HF (LVEF ≤ 35%) participated in a 12-week aerobic training program consisting of supervised sessions three times per week followed by a home-based phase. The intervention led to modest but statistically significant improvements in VO_2_peak of +0.6 mL·kg^−1^·min^−1^ after 3 months and +0.7 mL·kg^−1^·min^−1^ after 12 months. In contrast, a 16-week MCT program in older patients (≥60 years, LVEF ≈ 31%) did not improve mean VO_2_peak, although 26% achieved gains ≥ 10% [32].

Casillas et al. reported that VO_2_peak increased significantly only in the concentric endurance-training group over a 7-week period, rising by 2.0 mL·kg^−1^·min^−1^ [36]. Similarly, Alves et al. found a VO_2_peak increase of +3.8 mL·kg^−1^·min^−1^ after 12 weeks of moderate endurance training in HFrEF patients with permanent atrial fibrillation (AF) [41].

**Interval training**: Overall, HIIT tended to produce the largest short-term VO_2_peak gains (≈+2 to +3.5 mL·kg^−1^·min^−1^).

Ellingsen et al. [28] compared HIIT, MCT, and regular exercise recommendations over 12 weeks, reporting VO_2_peak changes of +1.4, +0.8, and −1.0 mL·kg^−1^·min^−1^, respectively, with effects not maintained at 1-year follow-up. In other 12-week programs, Huang et al. [30] found a +2.2 mL·kg^−1^·min^−1^ increase with modified HIIT, and Fu et al. [31] observed a gain of ≈+2.6 mL·kg^−1^·min^−1^ with interval training at 40% and 80% of VO_2_peak. Fernandes-Silva et al. [34] reported larger improvements of +3.5 mL·kg^−1^·min^−1^ in patients with low inflammatory biomarkers, and Alshamari et al. [35] showed VO_2_peak increases of +3.1 mL·kg^−1^·min^−1^ with HIIT and +1.6 mL·kg^−1^·min^−1^ with HIIT plus strength training.

In patients receiving CRT, HIIT increased VO_2_peak by +1.7 mL·kg^−1^·min^−1^ compared with +0.6 mL·kg^−1^·min^−1^ in the CRT-only group [38]. Both HIIT and MCT improved VO_2_peak in the study by Sales et al., accompanied by reductions in muscle sympathetic nerve activity [39].

**Combined aerobic and resistance and multi-component programs**: Overall, combined aerobic-resistance programs yielded improvements comparable to those achieved with HIIT (≈+1.5 to +3 mL·kg^−1^·min^−1^).

Rengo et al. [33] reported a +2.0 mL·kg^−1^·min^−1^ increase in VO_2_peak after 36 sessions of combined endurance and strength training. Similarly, Antunes-Correa et al. [37] found that a 16-week cycling and strength exercise program led to a +2.7 mL·kg^−1^·min^−1^ gain in VO_2_peak.

Fabri et al. [40] observed significant aerobic improvements after 12 weeks of supervised combined training, corresponding to an increase of ≈+2 METs (≈+7 mL·kg^−1^·min^−1^). Guimarães et al. [42] reported a VO_2_peak rise from +1.6 mL·kg^−1^·min^−1^ following 12 weeks of combined endurance and resistance training.

In a randomized study by Andrade et al. [43], moderate endurance and resistance training over 12 weeks increased VO_2_peak by +2.7 mL·kg^−1^·min^−1^ in the center-based group compared with +0.8 mL·kg^−1^·min^−1^ in the home-based group. The smallest trial, by Giuliano et al. [44], reported a VO_2_peak increase of +2.4 mL·kg^−1^·min^−1^ after 4 weeks of low-load, high-repetition exercises followed by 4 weeks of combined training, compared with only +0.2 mL·kg^−1^·min^−1^ after 8 weeks of combined training alone.

#### 3.3.2. Ventilatory Efficiency (VE/VCO_2_ Slope)

Six studies evaluated changes in the VE/VCO_2_ slope; four reported a significant reduction, with an average decrease of approximately 4.4 units, and one study showed a non-significant trend toward improvement.

Huang et al. [30] reported a decrease in the VE/VCO_2_ slope from 32.4 to 30.0 after 12 weeks of moderate HIIT. Alshamari et al. [35] found a non-significant trend toward improvement following HIIT. Alves et al. [41] reported a decrease from 38.5 to 32.1 after 12 weeks of endurance training in HFrEF patients with permanent AF.

Antunes-Correa et al. [37] observed decreases from 38.1 to 34.4 in men and from 40.0 to 35.4 in women after four months of combined endurance and strength training. Guimarães et al. [42] also found a significant decrease after 12 weeks of combined training, whereas Andrade et al. [43] did not observe a significant change in a smaller sample of 23 patients after a similar intervention lasting 12 weeks.

#### 3.3.3. Quality of Life (QoL)

Eleven studies assessed QoL using validated questionnaires, mainly the Minnesota Living with Heart Failure Questionnaire (MLHFQ) [29,31,32,35,37,41,43], the Kansas City Cardiomyopathy Questionnaire (KCCQ) [27], and the SF-36 or related scales [33,40]. Nine studies reported statistically significant within-group improvements, averaging a reduction of approximately 5 points in MLHFQ scores (Table 1).

**Aerobic endurance training**: In the HF-ACTION trial [27], aerobic exercise training was associated with a KCCQ increase of +5.2 points in the exercise group, compared to 3.3 in controls, with improvements sustained over 2.5 years. Brubaker et al. [32], which included older patients, found no significant group difference, as the control group improved more (MLHFQ −6 vs. −4.6). Alves et al. 2022 [41] reported a 16-point MLHFQ reduction after 12 weeks of endurance training in patients with AF.

**Interval training**: Fu et al. [31] reported significant QoL improvements after 12 weeks of aerobic interval training, with MLHFQ decreasing by 18 points, and SF-36 Physical and Mental scores increasing by 9 and 12 points, respectively. Alshamari et al. [35] found significant MLHFQ reductions after 12 weeks, −12 points with HIIT and −11 points with HIIT plus strength training. Santa-Clara et al. [38] observed improved QoL after CRT implantation in both groups, without additional benefit of HIIT over usual care.

**Combined training**: Dalal et al. (REACH-HF) [29] reported a mean reduction of −5.7 in MLHFQ after 12 weeks of home-based aerobic training. Rengo et al. [33] found that SF-36 scores increased from 57 to 69 and depressive symptoms (PHQ-9) decreased from 5 to 3. Fabri et al. [40] reported improvement in six or more SF-36 domains after combined training. Andrade et al. [43] observed a reduction in MLHFQ from 35 to 22 after 12 weeks of center-based exercise with no change in the home-based group.

#### 3.3.4. Mortality and Hospitalizations

HF-ACTION was the only trial powered for clinical events. Unadjusted analyses showed a non-significant trend toward reduced mortality and hospitalization with exercise training. However, when models were adjusted for key baseline predictors of prognosis—including exercise capacity, LVEF, depressive symptoms, and atrial arrhythmias—exercise training was associated with modest but statistically significant reductions in all-cause mortality or hospitalization (HR 0.89) and cardiovascular mortality or HF hospitalization (HR 0.85) [26].

#### 3.3.5. Additional Findings

-**Home-based vs. center-based training**: Both groups improved VO_2_peak, with the center-based group showing larger increases (+2.7 vs. +0.8 mL·kg^−1^·min^−1^) and greater QoL improved (MLHFQ −13 vs. −1) [43].-**Low-mass, high-repetition****training**: The PRIME protocol increased VO_2_peak by +2.4 mL·kg^−1^·min^−1^, whereas combined training alone improved VO_2_peak by only +0.2 mL·kg^−1^·min^−1^ [44].-**Inflammatory Biomarkers**: Improvements in VO_2_peak (+3.5 mL·kg^−1^·min^−1^) only in low-inflammation groups [34].-**Muscle and Vascular Function**: Exercise decreased muscle sympathetic nerve activity, improved forearm blood flow, and reduced vascular resistance; HIIT induced larger effects than MCT [37,39,42].-**NYHA class**: Dyspnea improved by 0.8 points after 4 months of supervised training in both men and women, with no significant change in untrained controls [37].-**Echocardiography**: LVEF improved in some studies, including 39% to 44% with supervised combined training and 31% to 36% in patients with permanent AF [40,41], while older adults showed no change [32]. HIIT and MCT reduced left ventricular end-diastolic diameter (LVEDD, −2.8 mm and −1.2 mm, respectively) [28]. Left atrial dimensions also decreased in HFrEF patients with permanent AF [41].-**In older patients (≥60 years)**: VO_2_peak did not improve significantly overall (mean change −0.2 mL·kg^−1^·min^−1^), although 26% of participants achieved ≥ 10% individual improvements [32].-**Patients with HFrEF and permanent AF**: Aerobic training increased VO_2_peak by +3.8 mL·kg^−1^·min^−1^, along with improvements in HR and QoL [41].-**Post-cardiac resynchronization therapy (CRT)**: CRT combined with HIIT, as well as HIIT alone, improved exercise capacity, QoL, and LVEF [38].

## 4. Discussion

### 4.1. Overview

Across the included studies, structured exercise training consistently improved aerobic capacity, ventilatory efficiency, and health-related QoL in patients with HFrEF. Mechanistic patterns observed across trials help explain these benefits. Subgroup differences, including attenuated responses in older adults or those with elevated inflammation and larger gains in patients with permanent AF, further highlight how individual patient characteristics influence training responsiveness.

### 4.2. Aerobic Capacity (VO_2_peak)

Most trials (16 of 17) showed meaningful increases in VO_2_peak, typically around +2 mL·kg^−1^·min^−1^ after 8–24 weeks of structured exercise training [26,28,30,31,33,34,35,36,37,39,40,41,42,43,44]. This magnitude is clinically relevant, as VO_2_peak is a strong prognostic marker in HF and even a 1 mL·kg^−1^·min^−1^ increase has been associated with a 10–15% reduction in mortality risk [45,46,47,48]. Small absolute gains may be particularly important for patients with values below the 12–14 mL·kg^−1^·min^−1^ threshold commonly used to identify advanced HF or transplant candidacy [49]. A 2 mL·kg^−1^·min^−1^ improvement corresponds to roughly 0.6 METs, a change known to carry meaningful reductions in long-term mortality risk [18,19].

However, improvements were not uniform across patient groups. In older patients (≥60 years), only a subset (26%) achieved a ≥10% increase in VO_2_peak, indicating a heterogeneous response to exercise training [32]. This attenuated effect, particularly beyond age 70, is well described in the literature and reflects age-related limitations in cardiac output, peripheral oxygen extraction, and overall physiological adaptability [50,51]. Evidence in older adults shows that endurance training produces the largest gains in VO_2_peak, with combined aerobic-resistance programs also providing meaningful improvements. Accordingly, individualized or multimodal training approaches that incorporate resistance work are recommended to optimize adaptations in this population [52].

Similarly, patients with high inflammatory markers showed no improvement [34]. Elevated inflammation is associated with blunted training responsiveness, including smaller gains in aerobic capacity and muscle remodeling. Chronic low-grade inflammation impairs muscle function and limits physiological adaptation, although exercise—particularly aerobic or combined aerobic-resistance training—can still reduce systemic inflammatory markers [53,54].

Notably, the largest increase was observed in the HFrEF group with permanent AF (+3.8 mL·kg^−1^·min^−1^) [41]. Exercise training has been shown to significantly improve cardiopulmonary function in AF, particularly when training intensity exceeds 50%. These adaptations—such as greater stroke volume, increased cardiac output, enhanced myocardial perfusion, and more efficient oxygen transport and utilization—may help explain the pronounced VO_2_peak response in this group [55].

### 4.3. Ventilatory Efficiency—VE/VCO_2_ Slope

The VE/VCO_2_ slope is a robust prognostic marker in HF with steeper slopes (>34–36) consistently associated with higher mortality and hospitalization risk, and each 1-unit increase is linked to a 4–10% rise in adverse events [21,22,49,56,57,58,59]. Conversely, reductions in VE/VCO_2_ reflect improved ventilatory efficiency and are associated with better survival [58]. Across the included trials, exercise training reduced the VE/VCO_2_ slope by approximately 4.4 units [30,37,41,42,43], a change considered clinically meaningful. These improvements likely arise from enhanced cardiac output and pulmonary perfusion—reducing ventilation–perfusion mismatch and physiological dead space—as well as attenuation of excessive ventilatory drive mediated by heightened chemo- and ergoreflex activation. Improved skeletal-muscle metabolism further lowers afferent signaling and delays early lactic acidosis, collectively reducing the ventilatory requirement for a given CO_2_ output [60].

Moderate-intensity continuous training, HIIT, and combined endurance-strength programs can enhance ventilatory efficiency. Variability in VE/VCO_2_ responses may largely reflect the small sample sizes of several trials, which limits statistical power, rather than true differences between training modalities. Notably, the largest improvement was observed in HFrEF patients with permanent AF (−6.4 units) [41], mirroring the disproportionately high VO_2_peak gains seen in this subgroup. Given that AF impairs cardiopulmonary coupling through irregular ventricular filling, these patients may derive greater benefit from training via increases in cardiac output and improved rate control [55]. The absence of significant change reported by Andrade et al. [43] may relate more to the small sample size (*n* = 23) than to the shorter training duration of three 30-min sessions per week over 12 weeks.

### 4.4. Quality of Life (QoL)

Across the 11 studies assessing QoL, exercise training consistently improved health-related QoL in patients with HFrEF. Most trials demonstrated significantly greater gains in the exercise groups compared with controls, with average improvements of approximately 5 points in MLHFQ scores. A 5-point change is widely accepted in clinical research and practice as the benchmark for a clinically relevant difference in MLHFQ scores [61]. Benefits typically appeared early during training and were sustained when programs incorporated a structured transition to home-based exercise [27].

Training setting and modality influenced the magnitude of improvement. Center-based CR produced the largest QoL gains, likely due to enhanced supervision, structured progression, and social interaction, whereas home-based programs also improved QoL but to a lesser extent [43]. HIIT-based interventions tended to yield some of the most pronounced improvements (−12 to −18 MLHFQ points) [31,35].

Two studies did not show superior QoL benefits of exercise over control. In patients recently receiving CRT, QoL improved in both groups, likely because CRT itself produces substantial symptomatic relief by improving cardiac output and reducing ventricular dissynchrony, thereby overshadowing any additional benefit of exercise [38]. In older adults, both groups experienced QoL improvements, which may reflect optimization of usual care, increased clinical contact, and nonspecific participation effects. Notably, this was also the only study that failed to demonstrate a meaningful improvement in VO_2_peak, consistent with the well-described blunted physiological training response in older adults and helping explain the limited between-group differences in QoL [32].

Mechanistically, improvements in QoL appear to arise from several core components of structured exercise trainings. Gains in exercise capacity were strongly associated with improvements across emotional, social, and physical QoL domains, indicating that enhanced functional status—and better symptom control such as reduced dyspnea—play a central role [37]. The inpatient rehabilitation environment also provides intensified social support through continuous interaction with healthcare staff and peers, alongside temporary relief from home and caregiving responsibilities—factors that particularly benefit patients who enter rehabilitation with lower social or emotional well-being. In addition, the structured psychosocial and educational components of rehabilitation help reduce emotional distress and strengthen coping, further contributing to the observed QoL improvements [62].

### 4.5. Mortality and Hospitalization Rates

HF-ACTION remains the only large trial powered for clinical events. Although unadjusted results showed only a non-significant trend, adjusted analyses identified modest reductions in all-cause mortality or hospitalization (HR 0.89) and in cardiovascular mortality or HF hospitalization (HR 0.85). These findings suggest that improvements in functional capacity and ventilatory efficiency may translate into clinically meaningful reductions in morbidity, particularly in patients who maintain long-term adherence [26].

### 4.6. Additional Physiological Effects of Exercise Training

Exercise training elicited meaningful improvements in neurovascular regulation. Both HIIT and combined aerobic-resistance training reduced muscle sympathetic nerve activity and vascular resistance while enhancing peripheral blood flow [37,39,42]. These adaptations improve exercise tolerance and symptom burden. No consistent reductions in inflammatory biomarkers were observed [34], suggesting that anti-inflammatory effects may require longer interventions or be restricted to specific subgroups [53,54].

### 4.7. Cardiac Effects of Exercise Training

Several studies demonstrated favorable cardiac remodeling, including increases in LVEF [40,41], reductions in LVEDD [28], and decreased left atrial dimensions in patients with permanent AF [41]. These changes were not observed in older patients, consistent with reduced physiological plasticity in advanced age [32]. Overall, cardiac remodeling effects of exercise are modest but clinically meaningful when combined with optimized pharmacotherapy.

### 4.8. Mode of Exercise

HIIT and modified HIIT improved both VO_2_peak and ventilatory efficiency in several studies [28,30], although HIIT was not superior to MCT for altering left ventricular remodeling or aerobic capacity [28]. HIIT may be more effective in improving neuromuscular and vascular parameters [39]. When combined with resistance training, HIIT enhanced muscular strength, endurance, and anaerobic threshold [34]. High-intensity intermittent exercise tailored to individual capacity represents an efficient training option for many patients with HFrEF [63].

Low-mass, high-repetition strength training (PRIME-HF) also improved aerobic capacity in older patients and may represent a practical alternative exercise option for this population [44].

### 4.9. Supervised vs. Unsupervised Exercise and Cardiac Rehabilitation

Home-based training is safe and improves exercise capacity and QoL [29,43], but supervised programs more effectively enhance physical capacity, exercise tolerance, and QoL [28,40,42]. Supervision may improve adherence and ensure adequate training intensity, both of which are strong predictors of benefit.

Long-term maintenance remains challenging, underscoring the importance of transitioning from supervised to self-managed exercise [28]. HF-ACTION demonstrated that hybrid models combining supervised initiation with structured home-based continuation can sustain improvements in VO_2_peak and QoL [26]. Home-based programs also offer cost-effectiveness advantages [29].

Older adults derive substantial benefit from multidomain rehabilitation after acute cardiac decompensation, showing improvements in frailty, physical function, and QoL [64]. Improving access to rehabilitation through systematic in-hospital referral and close outpatient follow-up may increase participation rates [33].

### 4.10. Subgroup Variability

Exercise responses varied across subgroups. Younger patients and those with lower baseline inflammatory status experienced greater gains, whereas older adults exhibited more heterogeneous responses, potentially due to age-related physiological constraints or insufficient training intensity [32,34,50]. HFrEF patients with permanent AF or those post-CRT showed significant improvements in aerobic capacity, cardiac function, and QoL, although HIIT did not provide additional QoL benefit compared with CRT alone [38,41]. Further research is required to define optimal training regimens for these subgroups.

### 4.11. Clinical Implications

These findings support integrating structured exercise training into standard HFrEF care. A supervised period of combined endurance and strength training, followed by a transition to self-managed exercise, appears to offer the most sustainable benefits.

### 4.12. Comparison with Previous Reviews

A 2022 meta-analysis evaluating exercise across both HF phenotypes reported significant improvements in cardiorespiratory fitness, QoL, and selected cardiac parameters [25]. A 2024 network meta-analysis of 82 trials identified high-intensity aerobic interval and MCT as the most effective modalities for improving VO_2_peak, LVEF, and QoL, with combined aerobic-resistance programs offering additional but smaller benefits [65].

The present review builds on previous work by focusing exclusively on patients with HFrEF and by restricting the analysis to four clinically robust outcomes—VO_2_peak, VE/VCO_2_ slope, QoL, and cardiovascular events. It incorporates studies published through 2024, including modern HIIT protocols, combined training models, and trials involving distinct patient subgroups. In contrast to quantitative meta-analyses, this review provides a structured and transparent synthesis of evidence across multiple training modalities and patient cohorts, linking physiological mechanisms to clinically meaningful endpoints. This approach offers an updated and practical perspective that may support more individualized exercise prescription in contemporary HFrEF management.

### 4.13. Limitations

Several limitations should be acknowledged. Many included studies had small sample sizes, short intervention and follow-up periods, and substantial heterogeneity in exercise modality, training intensity, and outcome definitions, which limits comparability across trials. As a structured narrative review, this work did not involve dual independent screening, a formal risk-of-bias assessment, or meta-analytic pooling, and the findings should therefore be interpreted qualitatively. The literature search was conducted systematically across major databases, although the grey literature was not searched, and only English-language publications were included. In addition, this review focused on contemporary studies reporting VO_2_peak, ventilatory efficiency, QoL, and clinical events, and did not attempt to re-evaluate all studies examined in earlier meta-analyses. These methodological considerations should be taken into account when interpreting the results.

### 4.14. Future Research

Future studies should investigate the long-term effects of exercise interventions, identify optimal training regimens for diverse patient subgroups, and clarify predictors of responsiveness. Large, multicenter trials comparing HIIT, MCT, and combined modalities- with extended follow-up would help refine exercise prescription. Personalized approaches incorporating age, comorbidity burden, inflammatory status, and functional capacity may enhance the effectiveness of training programs.

## 5. Conclusions

Structured exercise training provides consistent improvements in VO_2_peak, ventilatory efficiency, and QoL in patients with HFrEF, supporting its integration into standard care as recommended by current guidelines. Training responses vary across subgroups: older adults and individuals with elevated inflammation show attenuated gains, highlighting the need for optimized protocols and strategies to address inflammation, whereas patients with permanent AF demonstrate substantial potential for improvement.

Further research should clarify which patient characteristics predict improvements in ventilatory efficiency and how different training modalities influence VE/VCO_2_ outcomes. An effective transition from supervised to home-based training is essential, with supervised programs particularly beneficial for patients with reduced well-being or poorer baseline QoL. Continued refinement of individualized exercise prescriptions will help maximize the therapeutic impact of structured exercise in HFrEF.

## Figures and Tables

**Figure 1 jfmk-10-00483-f001:**
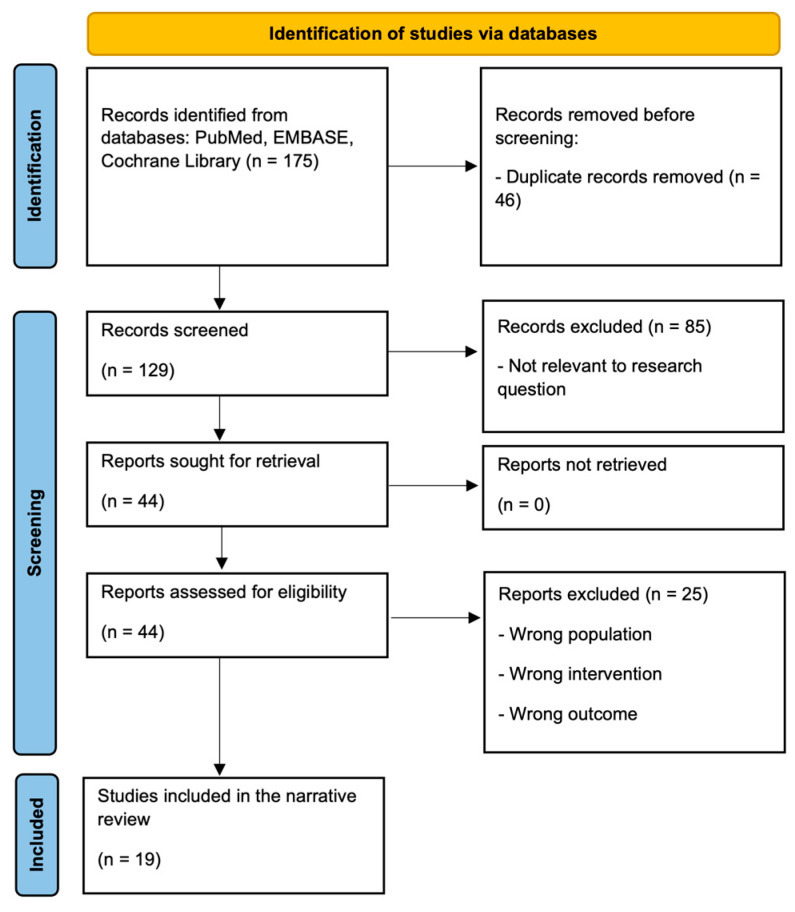
Flow diagram of study selection.

**Table 1 jfmk-10-00483-t001:** Clinical Trials on Exercise in Patients with HFrEF (Significant results in **bold**, non-significant results in *italics*).

Study (Year)	Participants (*n*)/LVEF	Design/Comparator	Intervention (Type/Duration)	VO_2_peak Change (mL·kg^−1^·min^−1^)	VE/VCO_2_ Slope Change	QoL Change (In Points)	Key Comments
O’Connor et al. (**HF-ACTION**) 2009 [26] and Flynn et al., 2009 [27] (Sub analysis)	2331/ ≤35%	RCT/usual care	Aerobic training/3×/wk 12 wk supervised → home 5×/wk	**+0.6 (3 months) +0.7 (12 months)**	-	**+1.93 KCCQ** (sustained for 2.5 years)	Reduced adjusted all-cause mortality or hospitalization (HR 0.89) and cv mortality or HF hospitalization (HR 0.85).
Ellingssen et al., 2017 [28]	261/ ≤35%	RCT/HIIT vs. MCT vs. RRE	HIIT or MCT or RRE/3×/wk 12 wk	**HIIT + 1.4** **MCT + 0.8** *RRE* *−* *1.0*	-	-	Effect not maintained at 1 year.
Dalal et al., 2019 (REACH-HF) [29]	216/ ≤45%	RCT/usual care	Home-based (chair-based and walking) 3×/wk 12 wk	-	-	**MLHFQ** **−5.7**	
Huang et al., 2014 [30]	68/ ≤40%	Prospective/mHIT vs. UC	mHIT 3×/wk 12 wk.	**+2.2**	**−2.4 (32.4** **→ 30.0)**	-	
Fu et al. [31]	60/ ≤30%	RCT/usual care	Intervals at 40% and 80% VO_2_peak/3×/wk 12 wk	**+~2.6**	-	**MLHFQ** **−18** **SF-36 Physical + 9** **SF-36 mental + 12**	
Brubaker et al., 2009 [32]	59/ ≤45% (age ≥ 60 yr)	RCT/usual care	Moderate endurance training/3×/wk 16 wk	*−0.2*	-	*MLHFQ* *−4.6*	A subset (26%) showed an increase in VO_2_peak by 10% or more.
Rengo et al., 2018 [33]	49/ ≤35%	Cohort- observational study/vs. baseline	Endurance and strength/36 CR sessions	**+2.0**	-	**SF-36 Physical** **+12** **PHQ-9 − 2**	**+1.6METs.** Only 11 completed all 36 sessions and exit measures
Fernandes-Silva et al., 2017 [34]	44/ ≤40%	RCT/usual care	Endurance (interval between VT_1_ and RCP)/3×/wk 12 wk	**+3.5 (low inflammation)** *No significant improvement (high inflammation)*	-	-	*No significant biomarker changes.*
Alshamari et al., 2023 [35]	44/ <50%	RCT/HIIT vs. COM	HIIT vs. COM (HIIT and strength)/3×/wk/12 wk	**+3.1 (HIIT)** **+1.6 (COM)**	*trend towards improvement*	**MLHFQ:** **−12 (HIIT)** **−11 (COM)**	Patients with LVEF < 50% included (but median LVEF 30%).
Casillas et al., 2016 [36]	42/ ≤45%	RCT/ECC vs. CON	Endurance (ergocyle vs. conventional cycle)/3×/wk 7 wk	*+1.9 (ECC)* **+2.0 (CON)**	-	-	**Triceps Surae Strength:** **+23% in ECC group.**
Antunes-Correa et al., 2010 [37]	40/ ≤40%	RCT/usual care	Endurance and strength/3×/wk 16 wk	**+~2.7 (both genders)**	**−3.7 (men)** **−4.9 (women)**	**NYHA:** **−0.8 (men)** **−0.7 (women)**	**FBF increased, vascular resistance** **decreased.**
Santa-Clara et al., 2019 [38]	37/ ≤40% (Patients with CRT)	RCT/HIIT and CRT vs. CRT alone	HIIT/2×/wk 24 wk	**+1.7** vs. +0.6	-	HeartQoL: +0.9 vs. +1.0; NHYA −1.2 vs. −1.1	Similar improvements in LVEF in both groups
Sales et al., 2020 [39]	30/ ≤40%	RCT/HIIT vs. MICT vs. usual care	HIIT vs. MICT/3×/wk 12 wk	**Significant increase** **(in HIIT** and **MICT)**	-	-	HIIT > MICT **significantly** **decreased MSNA**
Fabri et al., 2019 [40]	28/ <50%	RCT/usual care	Moderate endurance and resistance training/3×/wk 12 wk	**~7** (2 METs; only indirectly measured with METs)	-	**SF-36:** **improved in** ≥ **6 domains**	**LVEF 39% → 44%** (vs. 35% → 34% in the nontrained group)
Alves et al., 2022 [41]	26 ≤40% (permanent AF)	RCT/usual care	Moderate endurance/3×/wk 12 wk	**+3.8**	**−6.4**	**MLHFQ:** **−16**	**LVEF 31%** **→ 36%. Left atrial dimension 52** **→ 47 mm**
Guimaraes et al., 2021 [42]	24/ ≤40%	RCT/usual care	Endurance and resistance training 3×/wk 12 wk	**+1.6**	**Significant decrease**	-	**MSNA decreased, FBF increased**.
Andrade et al., 2021 [43]	23/ ≤40%	RCT/CR: home-based vs. center-based)	Moderate endurance and resistance training/3×/wk 12 wk	**+2.7 (center-based)** **+0.8 (home-based)**	*No significant changes in either group.*	MLHFQ: −13 (center-based) −1 (home-based)	
Giuliano et al., 2020 [44]	19/ ≤40% (Age ≥ 65 yr)	RCT/PRIME AND COMBO vs. COMBO	PRIME (8 strength exercises, 5 min each) vs. COMBO (endurance and strength training)/ 2×/wk 8 wk	**+2.4 (PRIME)** *+0.2 (COMBO)*	-	-	

Abbreviations: AF, atrial fibrillation; COM, combined training; COMBO, combined endurance-resistance training; CON, concentric training; CR, cardiac rehabilitation; CRT, cardiac resynchronization therapy; ECC, eccentric training; FBF, forearm blood flow; HeartQoL, Heart Quality of Life; HIIT, high-intensity interval training; KCCQ, Kansas City Cardiomyopathy Questionnaire; LVEF, left ventricular ejection fraction; MCT, moderate-continuous training; mHIT, modified high-intensity interval training; MLHFQ, Minnesota Living with Heart Failure Questionnaire; MSNA, muscle sympathetic nerve activity; NYHA, New York Heart Association; PRIME, Progressive Resistance Intensity Modulated Exercise; RCP, respiratory compensation point; RCT, randomized controlled trial; RRE, regular exercise recommendations; SF-36, Short Form 36 Health Survey; UC, usual care; VE/VCO_2_, ventilatory equivalent for carbon dioxide; VO_2_peak, peak oxygen uptake; VT_1_, ventilatory threshold 1; wk, week; yr, year.

## Data Availability

No new data were created or analyzed in this study. Data sharing is not applicable to this article.

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
