# Peer review of "Effects of Endurance and Resistance Training on Cardiovascular Outcomes and Quality of Life in Patients with Heart Failure with Reduced Ejection Fraction: A Structured Narrative Review"

_jfmk, 2025, doi:10.3390/jfmk10040483_

Round 1
Reviewer 1 Report
Comments and Suggestions for Authors
The manuscript presents an updated synthesis of exercise training in HFrEF. It is a clinically relevant topic, however it does not clearly define its novel contribution relative to prior systematic reviews and meta-analyses. Methods lack key PRISMA elements (registration, bias assessment, detailed search strategy).
The Discussion occasionally repeats findings rather than interpreting them critically. Major revisions are required to clarify the knowledge gap, strengthen methodological transparency, and improve organization and writing consistency.
Below you can find a detailed revision:
Abstract:
In the sentence: “Exercise is a promising intervention due to its potential to improve these outcomes…” the use of “promising” and “potential” suggests that exercise is experimental or not yet fully established, which is no longer accurate in the context of HFrEF.
As stated in the introduction, structured exercise training is an evidence-based, guideline-recommended component of standard care for stable HFrEF patients. Both the 2021 ESC Heart Failure Guidelines and the 2022 AHA/ACC/HFSA guidelines explicitly recommend that patients with heart failure who are able to participate should engage in exercise training or regular physical activity to improve functional status, exercise performance, and quality of life.
Introduction:
The background provides clear epidemiological context, but it lacks the knowledge gap this review intends to address.
Could you state more explicitly why a new systematic review is needed despite existing meta-analyses.
The sentence “About 50% of HF patients have HFrEF, 48 and over 50% are female [1–3].” Seems inaccurate, HFrEF is typically more prevalent among men Please verify references [1–3] and correct accordingly. Moreover, Are you referring to europe or a specific country?
I suggest to add a bridge sentence that create a more logical flow between the distinction pharmacological and non-pharmacological treatment and the discussion on exercise
For example, something like among non-pharmacological strategies, structured exercise training represents the most evidence-based intervention for improving both physiological and patient-centered outcomes.”
Moreover you could cite here key clinical guidelines instead of at the end of the paragraph, before stating the aim you should explain the gap you want to address.
The section on mechanisms (oxygen transport, anaerobic glycolysis, etc.) is interesting but it can be summarized
The introduction lists the benefits of exercise (mortality, QoL, VO₂, VE/VCO₂), but it does not clarify what remains uncertain in the literature.
Are you aiming to examine the average magnitude of effect, differences between training modalities (HIIT vs. MCT), or comparative effects on distinct outcomes (VO₂ vs. QoL vs. mortality)?
Please add a sentence after the aim explaining the potential impact
Use abbreviations consistently: introduce “cardiorespiratory fitness (CRF)” once, then apply it uniformly throughout.
Methods
Include the complete Boolean search string in an appendix or supplementary file, indicating MeSH terms and additional filters used.
The search was limited to the last 20 years and English language only, please justify these restrictions. Were any potentially relevant non-English studies excluded after screening?
Please clarify whether grey literature (e.g., trial registries, conference proceedings) was considered. This affects risk of publication bias.
The date “October 6, 2024” should be highlighted as the final search date; specify if any update search was performed before submission.
Please expand Figure 1 or add a short table detailing the main reasons for exclusion with counts for each category, following the PRISMA 2020 flow-diagram format. Please add the standard section headers (“Identification,” “Screening,” “Eligibility,” “Included”)
Regarding the screening: how many reviewers independently screened titles/abstracts and full texts.
Was there dual independent screening with a consensus or third-reviewer resolution process?
Why was no quantitative synthesis attempted, given that many studies report VO₂peak changes in comparable units? Consider adding a brief explanation (e.g., variability in interventions, durations, and outcome metrics precluded meta-analysis).
Please indicate who performed data extraction and whether it was verified by a second reviewer.
Describe the data items collected (study design, sample size, intervention details, comparator, outcomes, follow-up length, statistical results, etc.).
Clarify whether data extraction followed a standardized template or checklist, and how discrepancies were resolved.
Indicate if any attempt was made to contact authors for missing or unclear data.
No information is provided on risk-of-bias or quality assessment. This is a critical omission in a systematic review.
Which tool (e.g., Cochrane RoB 2 for RCTs, Newcastle-Ottawa for non-randomized studies) was used, if any? If not performed, please justify and add this as a limitation in both Methods and discussion.
Have you limited the search to a specific type of research design?
A short description of how results were summarized (grouped by training type, outcome, or study design) would enhance clarity.
Results
The Results section is currently very long and reads as a mix of descriptive summary, interpretation, and discussion. It would benefit from clearer subheadings and a more concise synthesis. Consider grouping studies by training modality, outcomes, or study design rather than presenting long study-by-study summaries.
Statements such as “modest but statistically significant benefits,” “substantial improvements,” and “consistent improvements” constitute interpretation and should be moved to the Discussion. Results should remain objective and descriptive.
Many studies report VO₂peak change in ml/kg/min, yet the manuscript never provides even a simple range or median effect across studies. Since no meta-analysis was performed, at least provide a structured summary table or visual (e.g., forest-style descriptive plot) to allow readers to understand typical effect magnitude. (e.g., “Across 16 studies reporting VO₂peak, increases ranged from .... to ... ml/kg/min over 8–24 weeks.”). Currently, the reader must extract this manually from text.
It is unclear the study selection in - VO₂peak, VE/VCO₂, QoL Subsections The narrative focuses heavily on specific trials (HF-ACTION, Huang, Ellingsen, etc.), but the criteria for which studies receive emphasis is unclear. Why are some studies discussed at length and others only briefly?
The Results section does not acknowledge or summarize the heterogeneity, and
Table 1, while detailed, is not enough to help the reader understand how differences across studies affect interpretation.
Table 1 is very long and dense, making it hard for a reader to extract key patterns.
Please add a short synthesized paragraph at the beginning of the Results section describing heterogeneity, the range and variability in terms of sample characteristics, exercise intensity, supervision modality, clinical severity (LVEF, NYHA class, age) ect.
Discussion
Much of the Discussion restates numerical findings already described in the Results. The Discussion should focus on Why differences may exist, Clinical relevance, interpretation and where uncertainty remains
The Discussion needs to articulate more clearly what this review adds beyond existing meta-analyses. At present, the contribution appears descriptive rather than novel. For publication, the unique value must be clarified.
Please provide also more critical analysis on Why did older patient cohorts not improve VO₂peak consistently? Was intensity insufficient? Sarcopenia? HF-related muscle metabolism?
Why did patients with high inflammatory biomarkers not respond? Could inflammatory state be a biomarker of exercise responsiveness?
The Discussion notes that exercise should be integrated into care, but this is already guideline-established. The manuscript should instead address:
- Optimal supervision duration before transitioning to home exercise
- Which training types may suit specific patient profiles (older, AF, post-CRT, high inflammatory biomarkers)
Several methodological limitations were not acknowledged:
- Lack of systematic risk of bias assessment
- Restriction to English-language trials
- Heterogeneity of outcome measures
- Short-term follow-up in most included studies
- Non-registration of the review protocol (which reduces transparency)
Reviewer 2 Report
Comments and Suggestions for Authors
The manuscript entitled “Effects of Endurance and Resistance Training on Cardiovascular Outcomes and Quality of Life in Patients with Heart Failure with Reduced Ejection Fraction (HFrEF): A Systematic Review” was reviewed. The article provides interesting information on the topic; however, adjustments need to be made so that the article can continue the path to publication.
I kindly ask if all changes made to the text be highlighted in yellow or a different color in the text.
Below are the reviewer's considerations to be adjusted in the manuscript.
Title:
1- Avoid using abbreviations in the title. Please remove the acronym "HFrEF."
Abstract:
2- Why did you use only three databases for the review?
3- Regarding the timeline, systematic reviews should not be limited to a single period.
4- Try to present more accurate information about the results found in the selected articles in the abstract.
5- Try to conclude the abstract by focusing on addressing the proposed objectives more directly.
Keywords:
6- Avoid repeating words already mentioned in the title.
Introduction:
7- Check the manuscript's formatting. I believe that paragraphs in the introduction should begin with an indent. Also, check whether the rules allow for line breaks between paragraphs. Also, try to make paragraphs more consistent; some have only a few lines.
8- At the beginning of the introduction, include epidemiological information from around the world, not just from Europe. This will demonstrate that the issue has global implications.
9- In the second paragraph, present the main causes of DKA.
10- The third paragraph requires further clarification. Please provide more information about pharmacological treatments, such as the type of medication and why they are essential.
11- When presenting the exercise variables, please explain how a good VO2max impacts people's health and quality of life at the metabolic and molecular level.
12- The introduction does not contain the study question; please include it. 13- Could you clarify the reason for choosing the outcome variables for the search, such as why you didn't include muscle strength, for example?
14- The objective at the end of the introduction is different from that presented in the abstract. Please standardize.
Methods:
15- You repeat the objectives at the beginning of the methods section. Please delete them.
16- Why do you only use three databases?
17- Why did you restrict the search to articles in English and published in the last 20 years? It's worth noting that systematic reviews should not have these restrictions.
18- In the search, you included the word VO2 and only VO2peak in the inclusion criteria. Why did you do it this way and not include VO2max?
19- Why didn't you register the review?
20- The flowchart you used to represent the prism is not standard. Please adjust it to the conventional model (https://www.prisma-statement.org/prisma-2020-flow-diagram).
21- What types of studies were included in your research? Please indicate.
22- Did you use any scale to assess the quality of the selected articles? For example: PEDro or Newcastle Ottawa.
23- Please provide a detailed description of the search strategy for each database used.
Results:
24- Adjust the tables. They are in bold and do not contain captions for abbreviations.
25- Table 1 sometimes makes it unclear whether peak or maximum VO2 is being assessed. Please check and adjust.
26- VE/VCO₂ was not presented in the searches. Please check.
27- Regarding quality of life, why didn't you use the names of the instruments presented in Table 2 for the searches?
Discussion:
28- In the discussion, there's no need to create a thread just to present the objectives.
29- The discussion is very fragmented, and you're presenting items that weren't presented in the article searches. Please adjust.
30- Regarding the limitations presented at the end of the discussion, I believe you should rethink the reviewer's comments.
Conclusions:
31- Be careful to complete only the study objectives.
Appendix:
32- I believe that the information presented in the summary could be taken from here and included in the description of the results or form part of the discussion.
Reviewer 3 Report
Comments and Suggestions for Authors
Thank you for the opportunity to review the manuscript entitled Effects of Endurance and Resistance Training on Cardiovascular Outcomes and Quality of Life in Patients with Heart Failure with Reduced Ejection Fraction: A Structured Narrative Review.
As standardized reporting guidelines for narrative reviews (analogous to PRISMA for systematic reviews) are not currently available, such manuscripts rely primarily on clarity, methodological transparency, and coherence of interpretation. In this regard, the submitted manuscript meets the key expectations for a narrative review. The structure is logical, the scope is well-defined, and the discussion is generally balanced. One aspect that could be strengthened is the explicit presentation of the review’s key strengths, which would help readers better appreciate its methodological value and contribution to the field. Additionally, it may be worth considering whether the title should be refined to indicate that this work represents a mini structured narrative review, thereby providing readers with an immediate sense of its format and depth.
Author Response
Thank you for the opportunity to review the manuscript entitled Effects of Endurance and Resistance Training on Cardiovascular Outcomes and Quality of Life in Patients with Heart Failure with Reduced Ejection Fraction: A Structured Narrative Review.
As standardized reporting guidelines for narrative reviews (analogous to PRISMA for systematic reviews) are not currently available, such manuscripts rely primarily on clarity, methodological transparency, and coherence of interpretation. In this regard, the submitted manuscript meets the key expectations for a narrative review. The structure is logical, the scope is well-defined, and the discussion is generally balanced. One aspect that could be strengthened is the explicit presentation of the review’s key strengths, which would help readers better appreciate its methodological value and contribution to the field. Additionally, it may be worth considering whether the title should be refined to indicate that this work represents a mini structured narrative review, thereby providing readers with an immediate sense of its format and depth.
Dear Reviewer,
thank you very much for your thoughtful and constructive evaluation of our manuscript. We greatly appreciate the time and expertise you dedicated to the review.
We have addressed your suggestion by adding a concise description of the key strengths of our review in the revised Discussion (Section 4.12). In this paragraph, we highlight that our work focuses exclusively on HFrEF, restricts the analysis to four clinically robust outcomes (VO₂peak, VE/VCO₂ slope, quality of life, and cardiovascular events), and synthesizes evidence across different patient cohorts and training modalities using a structured and transparent approach. We believe this addition clarifies the methodological value and contribution of the review.
Regarding the suggestion to modify the title, we carefully considered this point. Because our literature search was comprehensive- covering three major databases using a predefined strategy- we felt that adding the term “mini” might not accurately represent the scope and rigor of the review. For this reason, we opted to retain the original title.
We sincerely appreciate your constructive feedback, which helped us improve the clarity and impact of the manuscript.
Round 2
Reviewer 2 Report
Comments and Suggestions for Authors
Dear authors,
Thank you for providing the revised version of the manuscript. After reviewing all the adjustments made to the text, it has been verified that the material has evolved even further compared to the first version. Therefore, my suggestion is that the article be accepted for publication.
Author Response
Dear authors,
Thank you for providing the revised version of the manuscript. After reviewing all the adjustments made to the text, it has been verified that the material has evolved even further compared to the first version. Therefore, my suggestion is that the article be accepted for publication.
Dear reviewer,
Thank you very much for your positive evaluation and supportive feedback. We greatly appreciate the time and effort you dedicated to reviewing our revised manuscript. Your insightful comments in the first review round helped us to improve the quality of our work substantially.